# Genistein Up-Regulates the Expression of EGF and E-Cadherin in the Treatment of Senile Vaginitis

**DOI:** 10.3390/molecules27082388

**Published:** 2022-04-07

**Authors:** Yarui Sun, Lei Wang, Bo Wang, Yanli Meng, Weiming Wang

**Affiliations:** Heilongjiang Academy of Chinese Medicine Sciences, Harbin 150040, China; syr163wyyx@163.com (Y.S.); wlbaofu@163.com (L.W.); wangbo860830@163.com (B.W.)

**Keywords:** genistein, senile vaginitis, ovariectomy, epidermal growth factor, E-cadherin

## Abstract

Investigating the therapeutic effect of genistein (Gen) on postmenopausal senile vaginitis (SV) and its mechanism of action. Adult SPF female Wistar rats were selected to establish a bilateral ovariectomized animal model (OVX), which simulated senile vaginitis dominated by estrogen deficiency in ovarian dysfunction. After 14 days of continuous treatment, the morphology of vaginal epithelial tissue was observed and various types of epithelial cells were counted, and the body mass and uterine and vaginal index of rats were measured. the levels of vaginal tissue secretion, microorganism, hormone and glycogen in each group were measured and the reproductive health was evaluated clinically. The protein expression and mRNA expression of epidermal growth factor (EGF) and E-cadherin (E-cadherin) in vaginal tissues were detected by immunohistochemistry and RT-PCR, respectively. Result showed that Genistein lowered vaginal pH, increased vaginal index and vaginal health score, thickened epithelial layers and improved vaginal tissue atrophy after administration. Genistein also increased the contents of glycogen and Lactobacillus in vagina, and promoted the expression of EGF, E-cadherin protein and mRNA. To sum up, there is no significant change in serum E2 and FSH levels, indicating that genistein has no effect on hormone levels in rats. genistein promoted the proliferation of vaginal epithelial cells, thickened epithelial layers and the vaginal wall, which improved the resistance of vaginal epithelium, the recovery of self-cleaning ability and healed the vaginal wound and erosive surface to improve atrophy.

## 1. Introduction

Senile vaginitis (SV), also known as atrophic vaginitis, is one of the most common clinical conditions of menopause. It means that the depletion of ovarian follicles leads to the decrease of endogenous estrogen secretion after menopause, the loss of estrogen-dependent cells and the gradual pathological changes of vagina and other estrogen-dependent tissues, including vaginal atrophy, thinning of vaginal mucosa, decreased elasticity of connective tissue, decreased defense ability, susceptibility to bacterial invasion, and lack of estrogen will affect vaginal secretion and vaginal pH value, and increase the incidence of vaginal inflammation [1]. Its main clinical symptoms are the increase of vaginal secretion, peculiar smell, vulvar itching, burning pain and so on. In severe cases, it may also cause senile endometritis, pelvic inflammation and even urinary system diseases [2]. At present, estrogen and antibiotics are commonly used to treat SV, but some estrogen treatments are related to the increased risk of breast cancer [3]. In addition, potential side effects of long-term use of estrogen include vaginal blood loss, perineal pain, breast cancer, and long-term use of antibiotics that can lead to imbalances in the vaginal flora [2]. Therefore, the search for alternative agents that may provide beneficial effects similar to estrogen while devoid of its adverse effects is warranted.

Cytologically, in the hypoestrogenic state, there is a reduction in the number of epithelial cell layers, vaginal keratinised and exfoliated epithelial cells, little or no in the surface or intermediate squamous cells, an increase in basal, parabasal cells and inflammatory cells. The degeneration of the collagen and elastin fibres in the underlying connective tissue leads to smaller tissue elasticity and more fragile mucosa [4]. Estrogen stimulates epithelial cells to produce glycogen, which is metabolized into lactic acid by lactic acid bacteria to maintain the vaginal pH in the acidic range. The combination of acidic pH and hydrogen peroxide helps to inhibit the reproduction of other potentially pathogenic microorganisms in the vaginal ecosystem and maintain the dominant position of lactic acid bacteria [2]. Decreased in lactobacilli leads to increases of the pH levels [5], which are susceptible to other pathogenic bacteria such as staphylococci, group B streptococci and coliforms. In summary, vaginal morphology, cellular level, glycogen level, type and number of microorganisms are relevant to senile vaginitis and can be used as indicators of efficacy.

Epidermal growth factor (EGF) is a peptide that has strong pro-divisive activity on epithelial cells of various tissue origins, which can promote cell proliferation and maturation, accelerate the replacement of senescent cells and collagen synthesis, and enhance the growth of granulation tissue including the migration and proliferation of tissue cells, the formation of granulation tissue, extracellular matrix deposition. In addition, it also promotes cell migration to the damaged area to accelerate wound healing. Through a series of signalling, EGF is involved in the proliferation and differentiation of various tissue cells including endothelial cells, mammary epithelial cells [6] and stromal cells [7]. The mucosal epithelial layer can be repaired and the vaginal wall thickened by promoting the proliferation of squamous epithelial cells in the vaginal tissue. Studies have also shown that EGF can directly affect ovarian endocrine function, which can be evidenced by its ability to regulate follicular steroid production in mice at physiological concentrations, and to inhibit estradiol (E2) and promote serum follicle-stimulating hormone (FSH) secretion in pituitary cells [8]. It also promotes the proliferation of mouse oocytes to impede the communication between the oocyte complexes of the oocytes, which relieves the inhibition of oocyte meiosis and thus accelerate the maturation of the oocytes themselves. EGF can regulate oocyte maturation independently or optimize oocyte development combined with other factors [9] and has been reported in various mammalian studies [10,11,12]. It has also been reported [13] that FSH can enhance the role of EGF in promoting oocyte maturation, which was not studied in depth in this experiment and left for further study.

Genistein (5,7-dihydroxy-3-(4-hydroxyphenyl) chromen-4-one, Gen) is a phytoestrogen and isoflavone mostly found in soy and soy-derived foods [14] and can also be found in other foods. Studies have shown that genistein has estrogenic effects on estrogen-deficient animals and postmenopausal women [15], which acts on estrogen receptors (ERs), ER alpha and beta in the classical genomic mechanism [16]. Currently, there are many directions of research on dye lignin, such as [17] and [18], which showed that the growth of MCF-7 breast cancer cells was stimulated by low concentrations of genistein (10^−8^–10^−6^ mol/L) and inhibited by high concentrations (>10^−5^ mol/L). Experimental [19] found that low concentrations of genistein (≤0.1 mg/mL; 0.0037 mmol/L) in vitro caused human uterine leiomyoma cell proliferation, while higher exposure levels (≥1 mg/mL; 0.037 mmol/L) had an inhibitory effect. According to several literature reviews and studies [20,21,22,23,24], genistein clearly shows a strong potential in the prevention of breast cancer. Studies [19] and [25] demonstrated that low concentrations of genistein caused proliferation of human uterine leiomyoma cells. A daily oral dose of 54 mg of genistein has been shown to be an effective alternative therapy for the management of endometrial hyperplasia in premenopausal women. This concept was further validated by experimental findings [26,27], which showed a negative association between soy intake and endometrial cancer risk, and between genistein and ovarian cancer risk. This study [28] also demonstrated that born female rats administrated genistein (40 mg/kg) in subcutaneous leading to increased uterine weight, early puberty and even permanent oestrus. In addition to anti-tumour, osteoblastic and anti-cancer abilities, genistein has also been suggested the antioxidant effects [29], active cardiovascular [30] and antiadipogenic [31] effects. There has been much research of goldfinch isoflavone in postmenopausal hormone replacement therapy to reduce the severity of menopausal symptoms. Although hormone replacement therapies are currently available, many of them may increase the risk of thromboembolism, cancer, stroke and other complications [32]. The use of genistein as a form of hormone replacement therapy is common [33]. A 2016 pooled analysis of 62 clinical trials in 6653 postmenopausal women found that phytoestrogen supplementation was significantly associated with a reduction in the number of daily hot flushes and general vaginal dryness [34]. Literature on the clinical use of phytoestrogens showed that more and more peri- and postmenopausal women used genistein to alleviate menopausal symptoms [35,36]. However, there are few studies on the therapeutic effects and mechanisms of action of genistein in senile menopausal vaginitis, and this paper is a study in that direction. As a phytoestrogen commonly found in food, breast milk and human serum, genistein studies is critical to both maintaining global health and promoting progress in women’s health. Genistein is known to have a palliative effect on post-menopausal menopausal symptoms [15], we have therefore previously concluded that Gen has a positive effect to the treatment of age-related vaginitis. We have also learned that Epidermal growth factor (EGF) has a biological function in promoting cell proliferation and tissue repair, and Epithelial calcium adhesin (E-cadherin) is a major molecule in maintaining epithelial cell polarity and intercellular adhesion. Therefore, we had presupposed before the experiment that genistein promotes the proliferation and adhesion of vaginal epithelial cells by enhancing the expression of EGF and E-cadherin, repairing the vaginal wall, thus improving vaginal pH, regulating glycogen and flora levels, which ultimately achieved a therapeutic effect in vaginitis. To assess these possibilities, we established bilateral ovariectomized rats model treated with or without genistein, and the vaginal tissue morphology and cellular distribution of the rats were microscopically observed, and the pH, glycogen levels and number of bacterial species in the vagina were measured to confirm the therapeutic effect of Gen on vaginitis. Polymerase chain reaction (PCR) and immunohistochemistry were then used to examine the expression of EGF and E-cadherin in vaginal tissues to confirm the mechanism of action. The aim of this study was to examine these aspects of the therapeutic effect of genistein on postmenopausal senile vaginitis and to investigate its mechanism of action.

## 2. Results

### 2.1. Effect of Genistein on the General Condition of Rats with Vaginitis

There was no significant difference in body weight between the groups before surgery. Between de-ovulation and administration, it was observed that the sham-operated group (Sham) had normal fur colour and body shape compared to the Sham group, whereas other groups showed coarse and lustrous coat colour, redness and swelling of the vaginal opening, increased discharge, frequent itchy movements, and significantly larger body shape, with significant increased in body weight (*p* < 0.05, Table 1), significantly higher pH (*p* < 0.01, Figure 1A). After 14 days of administration, the positive control group (Y) and the genistein group (Gen) showed slightly coarser fur, reduced redness and swelling, decreased discharge, reduced inflammatory conditions, occasional pruritic movements, increased body size but much less pronounced than in the model group, sig-nificantly lower pH than in the M group (*p* < 0.05, Figure 1A), and significantly higher vaginal health scores (*p* < 0.01, Figure 1B). the Gen group had a more pronounced effect than the Y group.

### 2.2. Effect on Vaginal Histomorphology and Cytology of Rats with Vaginitis

HE staining showed that the squamous epithelial layer in the vaginal tissue of Sham rats was structurally intact, with a single layer of columnar epithelium covering the endothelial surface and tightly connected. Reduction in the number of squamous epithelial layers and thinning of the vaginal wall in the M group. In contrast, the number of vaginal squamous epithelium, the number of mucosal epithelial layers and the thickness of the vaginal wall were increased in the dosed rats compared to the M rats, with the most pronounced increase in the Gen group (Figure 2).

### 2.3. Effect on Vaginal Cytology in Rats

Figure 3 shows that between de-ovulation and administration, the rats in the Sham group had an estrous phase with a large number of epithelial cells (including unkeratinized cells, incompletely keratinized cells and keratinized exfoliated cells) in the vagina, and almost no inflammatory cell infiltration seen (Figure 3A). in the remaining groups, the ovaries were successfully removed for modelling, and inflammatory cell infiltration was evident in the mucosal layer and submucosal layer or the whole layer of the vagina, and epithelial cells were significantly reduced (Figure 3B–E). In Figure 4, it can be observed that the number of inflammatory cells in the vagina of the rats in the Y and Gen groups was significantly reduced and the number of epithelial cells (including unkeratinized cells, incomplete keratinized cells and keratinized exfoliated cells) was significantly higher than that in the M group, and the effect of the Gen group was better than that of the Y group (*p* < 0.05, Figure 4F). In contrast, there was a small increase in the number of epithelial cells and a slight decrease in the number of inflammatory cells in the K group, which was considered as the stimulating effect of vaginal drug.

### 2.4. Effect on Vaginal Indices in Rats

As can be seen from Figure 5, the vaginal index of the rats in the M and K groups was significantly lower than that of the Sham group (*p* < 0.01), which indicated that the effect of ovarian removal was significant. The vaginal index of all the administered groups was increased in different degrees after treatment, and the atrophy was improved.

### 2.5. Effect on Hormone Levels in Rats

After ovarian debulking, serum levels of E2 were significantly lower (*p* < 0.01, Figure 6A) and FSH levels were significantly higher (*p* < 0.05, Figure 6B) in the M group compared to the Sham group, which indicated a trend of decreased estradiol (E2) and increased follicle-stimulating hormone (FSH) during menopause after successful modelling. Figure 6 shows that serum E2 levels were significantly higher in group Y than in group M after administration and Figure 6B showed that the serum FSH level of group Y was significantly lower than that of group M after administration, which considered that the positive control drug (Suppositories of Baofukang) achieved its therapeutic effect on atrophic vaginitis by regulating hormone levels. In contrast, there was no significant change in the K and Gen groups, which indicated that the genistein administration group did not contain hormones and did not affect the hormone levels of the rats themselves.

### 2.6. Effect on Glycogen Content in the Vagina of Rats

Normally, oestrogen stimulates the epithelial cells to produce glycogen that is converted to lactic acid by lactic acid bacteria in the vagina, which resulted in an acidic pH. Figure 7 shows that compared to the Sham group, glycogen levels were significantly reduced in the M group and Glycogen levels were increased in both the Y and Gen groups after dosing.

### 2.7. Effect on Microorganisms in the Vagina of Rats

As can be seen in Figure 8, observation of lactobacilli in vaginal secretions after Gram staining showed a significant decrease in vaginal lactobacilli in the M group after surgery compared to the Sham group, and a significant increase in lactobacilli in the Gen group after administration of the treatment.

### 2.8. Effects on EGF and E-cadherin Protein Expression in Rat Vaginal Tissues

The immunohistochemical results of rat vaginal tissue sections were scored according to the immunohistochemical scoring criteria, and the following conclusions were drawn. the Sham group had darker and larger staining (Figure 9A) with a score of 8.13, which was high positive and high protein expression. In the M group, the cells were lightly stained and small in area (Figure 9B) with a score of 2.92, which was low positive and a low protein expression. The cells in the K group were lightly stained and smaller in area (Figure 9C) with a score of 2.83, which was low positive and a lower protein expression. Group Y cells (Figure 9D) with a score of 5.58 were positive and a moderate protein expression. Cells in the Gen group stained darker and had a larger staining area (Figure 9E) with a score of 7.88, which was strongly positive with high protein expression. EGF protein expression in Sham group, Y group and Gen group were significantly different compared with the model group (*p* < 0.01, Figure 9F), and Gen group had better results than Y group.

The Sham group had darker and larger staining (Figure 10A) and high protein expression with a score of 7.61, which was high positive. In the M group, the cells were lightly stained and small in area (Figure 10B) with low protein expression, which was low positive with a score of 2.58. The cells in the K group were lightly stained and smaller in area (Figure 10C) with lower protein expression and a score of 2.08, which was low positive. Group Y cells (Figure 10D) with moderate protein expression and a score of 2.83, were low positive. Cells in the Gen group stained darker and had a larger staining area (Figure 10E) with high protein expression and a score of 6.38, which was strongly positive. Epithelial calcium adhesin (E-cadherin) protein expression in Sham group and Gen group were significantly different compared with the model group (*p* < 0.01, Figure 10F). In the Gen administration group, an increase in the intensity of E-cadherin on the plasma membrane of epithelial cells was observed. This indicates an increased expression of cell adhesion proteins between adjacent cells.

### 2.9. Effects on mRNA Expression of EGF and E-Cadherin in Vaginal Tissues

Table 2 shows that the mRNA expression of EGF and E-cadherin was significantly higher in the Sham, Y and Gen groups compared to the M group (*p* < 0.01), and the mRNA expression in the Gen group even exceeded that of the Sham group, which indicated that the expression of EGF and E-cadherin was reduced in the vagina of rats with menopausal vaginitis. however, administration of genistein promoted the effect of increased EGF and E-cadherin mRNA transcript expression.

## 3. Discussion

Senile vaginitis is a common disease in post-menopausal women, which presents clinical symptoms such as abnormal leucorrhoea, itching and burning of the vulva. The pathogenesis of senile vaginitis is based on a decrease in ovarian function and follicle number [37], which leads to a decrease in effective glycogen, a decrease in lactobacilli production, an increase in vaginal pH [5], an increase in bacterial diversity, an increase in inflammatory vaginal cells, a thinning of the squamous epithelium, a decrease in the number of mucosal epithelial layers and an atrophy of the vaginal wall [4].

To further determine the influence of Genistein on Senile vaginitis, we used multiple methods to detect study. HE stain was used to visualize the vaginal tissue structure and pH, glycogen and lactobacilli were conducted to detect the acidic environment in the vagina. Compared to M rats, there was an increase in the number of vaginal squamous epithelium, mucosal epithelial layers and the thickness of the vaginal wall in the Gen group. It can be observed that pH value and the number of inflammatory cells was significantly reduced in the Gen groups. Glycogen and lactobacilli were significantly increased in Gen groups compared to M rats. These results provide new proof of Genistein in the curing of Senile vaginitis.

Genistein is the main phytoestrogen found in soybeans and soy products and acts as a natural alternative to estrogen avoiding the side effects and potential risks associated with hormone therapy [38]. The therapeutic efficacy of genistein in senile vaginitis and further mechanism research would have social and economic value.

E-cadherin presents in epithelial cells and is the main molecule that maintains epithelial cell polarity and intercellular adhesion [39,40]. A decrease in E-cadherin protein resulted in a non-epithelial character in the epithelial cells with reduced cell-cell adhesion and disrupting the structural integrity of the epithelium. It stains strongly on normal epithelial cell membranes and weak in the cytoplasm [41]. EGF can promote cell proliferation to repair the mucosal epithelial layer and thicken the vaginal wall. RT-PCR results showed that the mRNA expression of EGF and E-cadherin in the tissues of the genistein administered group was significantly higher than that of the model group, which suggested that genistein can increase the transcriptional level of EGF and E-cadherin. The results of immunohistochemistry showed that the intensity of EGF and E-cadherin labelling was increased in the Gen group, which showed that genistein can enhanced the protein expression of EGF and E-cadherin. Thus, genistein exert the biological function to promote cell proliferation and tissue repair. In summary, genistein can promote the proliferation of vaginal epithelial cells by significantly increasing the expression of EGF and E-cadherin in the vaginal epithelium. It improved the distribution ratio of vaginal cells from predominantly inflammatory cells before treatment to predominantly epithelial cells after treatment. It also promoted the keratinisation of superficial vaginal cells resulting in an increase in vaginal squamous epithelial cells, an increase in the number of mucosal epithelial layers, and an increase in intercellular adhesion. The vaginal wall was thickened and atrophy was improved, which promoted the recovery of the vaginal epithelium’s resistance and self-cleaning ability and the healing of vaginal wounds and erosions. In this experiment, genistein could enhance the expression of EGF and E-cadherin to promote the proliferation of vaginal epithelial cells and repair the vaginal wall, which improved the symptoms of senile vaginitis and had a better therapeutic effect on senile vaginitis. There are few studies on this direction, and this experiment has enriched the theory of the drug action of genistein, broadened the therapeutic direction of senile vaginitis and provided ideas for future research.

## 4. Materials and Methods

### 4.1. Materials

The experimental group was given 0.2 g/capsule of genistein suppositories containing 3.375 mg of genistein, the exact concentration of dye lignin in the finished product was 1.69%, where the purity of the genistein extract was 100%. It was extracted by the preparation laboratory of Heilongjiang Academy Of Chinese Medicine Sciences (20210910), Heilongjiang, China. Analytical method: The mobile phase is methanol 0.05% phosphoric acid aqueous solution (70:30), flow rate 1.0 mL/min, injection volume0.01 mL, detection wavelength 262 nm, column temperature 30 °C. Concerning the analytical instrument used: High Performance Liquid Chromatograph (Shimadzu, Kyoto, Japan, SPD-M20A); Disc Filter φ200 (Zhejiang Wenbro Machinery Valve Co., Zhejiang, China); Booster electric stirrer JB50-D type (Shanghai Specimen Model Factory, Shanghai, China); Ultrasonic Cleaner KQ-800KDB (Kunshan Ultrasonic Instruments Co., Kunshan, China).

Positive control group dosing: Suppositories of Baofukang.

Blank control group dosing: Blank fugitives.

### 4.2. Chemicals

Immunohistochemistry kits (Beijing Zhongsugi Jinqiao Biotechnology Company, Beijing, China), DAB Chromogenic Kit (Thermo Fisher Scientific, Shanghai, China), RT-PCR kits (Cwbio, Jiangsu, China), Primer synthesis (Shanghai Bioengineering Co., Shanghai, China), Glycogen PAS staining kit (Beijing Solarbio Science and Technology Co., Ltd. Beijing, China), PBS buffer (Beijing Solarbio Science & Technology Co., Ltd. Beijing, China), Hematoxylin and eosin stain (Nanjing Jiancheng Bioengineering Institute, Nanjing, China).

### 4.3. Instruments

Electrophoresis instrument, Versa Doc 4000 mp gel imager (Bio-Rad, Hercules, CA, USA), Line Gene 9600 Real-Time PCR (Hangzhou Bioer Technology Co., Hangzhou, China), Multifunctional Enzyme Labeler (Tecan Trading AG, Männedorf, Switzerland), DP72 Fluorescence Microscope (Olympus Corporation, Tokyo, Japan), Model ST-16R Cryogenic High Speed Centrifuge (Thermo Fisher Scientific, Shanghai, China), Waters ACQUITYTM UPLC Chromatograph (Waters Corporation, Milford, MA, USA), AB SCEIX Triple-TOFTM 5600+ High Resolution Mass Spectrometer (AB SCIEX Corporation, Framingham, MA, USA).

### 4.4. Animals and Experimental Design

In this case, 55 adult female Wistar rats of SPF class 250–300 g were fed with standard rat food and water. Animals were allowed to drink ad libitum and maintained on a 12-h light/12-h darkness cycle.

Eight rats were randomly assigned to undergo sham debulking (Sham group) and the rest were established as a bilateral ovariectomy (OVX) animal model. 1% Pentobarbital sodium 40 mg/kg intraperitoneal injection for anaesthesia in rats. The dead rats were removed and the status of the rats was observed daily and the presence of discharge (blood) from the vaginal opening, redness and congestion of the vaginal mucosa were recorded. After recovery from surgery, the vaginal pH of the rats was measured for five consecutive days and vaginal smears were performed (method: a sterile cotton swab dipped in saline was slowly rotated and inserted into the vagina of the rats, gently removed with a few rotations and applied to a clean slide, fixed and dried in 95% ethanol), and routine haematoxylin eosin staining (HE staining) was performed to observe the keratinization and shedding of epithelial cells. The model was established when the rat’s vagina became congested, red and swollen with a thick secretion, which indicated that the ovaries had been completely removed. Each group of 8 successfully moulded rats was randomly divided into four groups namely: model control group (M); blank bolus excipient group (K); positive control (pau d’arco bolus) group (Y); and genistein administration group (Gen). Vaginal administration, equal volume of 1 capsule/0.2 g/each, was administered for 14 consecutive days.

### 4.5. Cell Observation Count of Vaginal Lavage Smears

Before and after the administration of the drug, the vaginal health score was performed on a scale of 1 to 4 with higher scores associated with less atrophy. pH, discharge, epithelial mucosa, wetness, vaginal folds and mucosal colour were used as indicators. Vaginal lavage smears and HE staining were performed to observe the morphology of exfoliated cells, and high magnification microscopy was used to count exfoliated epithelium, inflammatory cells and epithelial classification.

### 4.6. Intravaginal Microbiological Testing

After the last dose, the vaginal lavage was collected for culture and detection of lactobacilli; the amount of lactobacilli in the vaginal secretions was observed by staining with Gram stain.

### 4.7. ELISA Kit to Determine Sex Hormone Levels

24 h after the last dose, blood was collected from the abdominal aorta and the serum levels of Estradiol (E2) and Follicle-stimulating Hormone (FSH) were measured according to ELISA kits.

### 4.8. Organs of the Uterus and Vagina Index

After the animal was executed, the upper end of the urethra was taken at about 1 cm, and the skin and abdominal wall were cut longitudinally at about 3 cm. The uterus and vagina were peeled out and the blood was flushed out with saline. Excess water was blotted out with filter paper and the animal was weighed together with an electronic balance to calculate the organ coefficient and perform pathological histological observation with the formula: Organ index = organ weight (mg)/body weight (g).

### 4.9. Determination of Glycogen Content

The vagina was taken 0.5 cm fixed in 10% formalin, dehydrated and embedded, oxidized with oxidant, stained sequentially with Schiff and haematoxylin stains, and differentiated and returned to blue. PAS (Periodic Acid-Schiff’s reaction) staining was carried out to observe the glycogen content in the vaginal tissue.

### 4.10. Histomorphological Observation of the Vagina

Vaginal tissue specimens were fixed in 10% formalin for 2 weeks, dehydrated in an alcohol series, xylene transparent, wax dipped, embedded and cut into 0.005 mm thick slices for HE staining to observe vaginal morphology and thickness in rats.

### 4.11. Immunohistochemical Detection of EGF, E-Cadherin Protein Expression

Rat vaginal tissue was dipped in wax, embedded and sectioned. After deparaffinisation, the tissue was incubated with hydrogen peroxide to quench the endogenous peroxidase activity and heated to inactivate the endogenous enzymes and biotin. Tissues were incubated with antibodies against the epidermal growth factor (EGF) and epithelial calreticulin (E-cadherin) for 60 min at 37 °C. Reaction enhancers were added dropwise, secondary antibodies were incubated for 20 min, developed with 3,3-diaminobenzidine (DAB) for 5 min and finally counterstained with haematoxylin. The tissue was observed with a microscope and photographed. To assess protein expression, the intensity of DAB was analysed and scored using Image J (1.48, National Institutes of Health, Bethesda, MD, USA) [42].

Three randomly selected fields of view were photographed under a light microscope for scoring and cells showing brown signals were defined as immunohistochemically positive. The intensity of staining was categorised as: negative 0, low positive 1, moderate positive 2 and high positive 3. The percentage of stained areas was 0 for 0 to 5%, 1 for 6% to 25%, 2 for 26% to 49% and 3 for >50%. The final score was measured by multiplying the combined scores of the two. A total score of 0 is considered negative, 1 to 3 as low expression, 4 to 6 as moderate expression and 6 to 9 as high expression.

### 4.12. Real-Time PCR Assay to Detect the mRNA Expression of EGF and E-Cadherin

Total RNA from rat vaginal tissues of each group was collected and extracted with TRIzol reagent (Thermo Fisher Scientific, Shanghai, China). The concentration of RNA was detected by Multifunctional Enzyme Labeler and then total RNA was reverse transcribed into cDNA according to the steps of reverse transcription kit. Amplification conditions were: 95 °C for 10 min, 95 °C for 10 s, 60 °C for 30 s, 72 °C for 30 s, 40 cycles; lysis curve analysis: 95 °C for 10 s, step sampling, step temperature 0.5 °C. The relative mRNA expression of EGF and E-cadherin in vaginal tissues of rats in each group was calculated using RN-ACTB as an internal reference. The primer sequences are shown below:

Internal reference β-actin

F-TGATGACATCAAGAAGGTGGTGAAG,

R-TCCTTGGAGGCCATGTGGGCCAT

EGF

F-TGACTATGACGGTGGCTCCATCC

R-CCCAGTGTGTTTGTCGGCTATCC

E-cadherin

F-CCTACAATGCTGCCATCGCCTAC

R-GGGTAACTCTCTCGGTCCAGTCC

### 4.13. Data Analysis

Data were analysed using SPSS 19.0 statistical software (Chicago, IL, USA). Statistical analysis was performed using one-way analysis of variance (ANOVA) to analyse differences between groups, with the relevant data expressed as x¯ ± s. When variance A showed significant differences between means, independent samples Tukey’s test was used to compare the data between the two groups and to determine which means were significantly different (*p* < 0.05 was considered a statistically significant difference).

## 5. Conclusions

Genistein increases the expression of EGF and E-cadherin in vaginal epithelial cells, promotes the proliferation of vaginal epithelial cells, enhances intercellular adhesion, thickens the epithelial layer and improves atrophy.

## Figures and Tables

**Figure 1 molecules-27-02388-f001:**
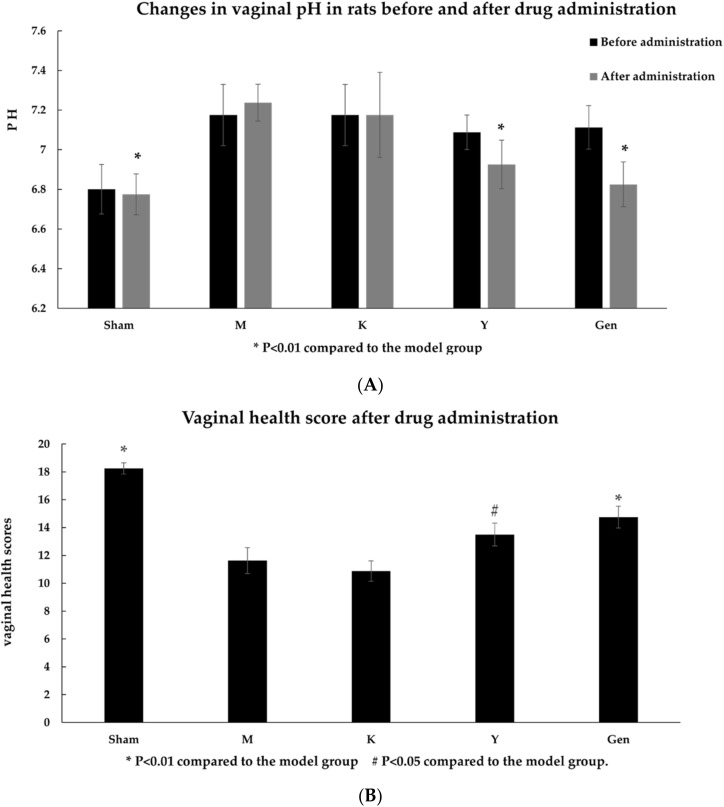
(**A**) Changes in vaginal pH in rats before and after drug administration; (**B**) Vaginal health score after drug administration. * *p* < 0.01 compared to the model group. # *p* < 0.05 compared to the model group.

**Figure 2 molecules-27-02388-f002:**
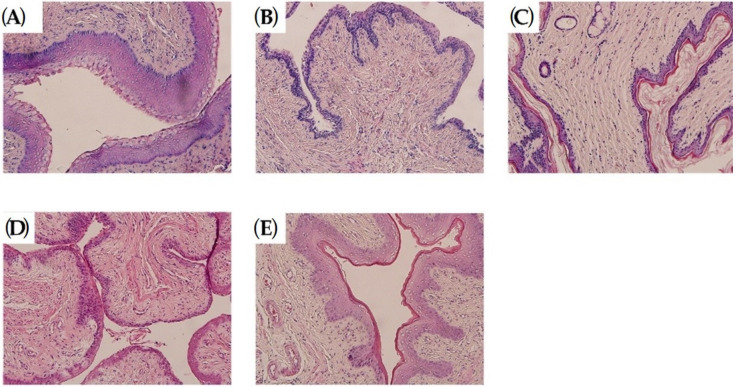
(**A**) HE stain of vagina of Sham group rats with normal morphology, (40Χ). (**B**) HE staining of the vagina of group M rats with significant atrophy of the vaginal wall, (40Χ). (**C**) HE staining of the vagina of rats in group K has not significantly different from those in group M, (40Χ). (**D**) HE staining of the vagina of rats in group Y showed a mild improvement in atrophy compared to group M, (40Χ). (**E**) HE staining of the vagina of rats in the Gen group showed a significant improvement in atrophy compared to the M group, (40Χ).

**Figure 3 molecules-27-02388-f003:**
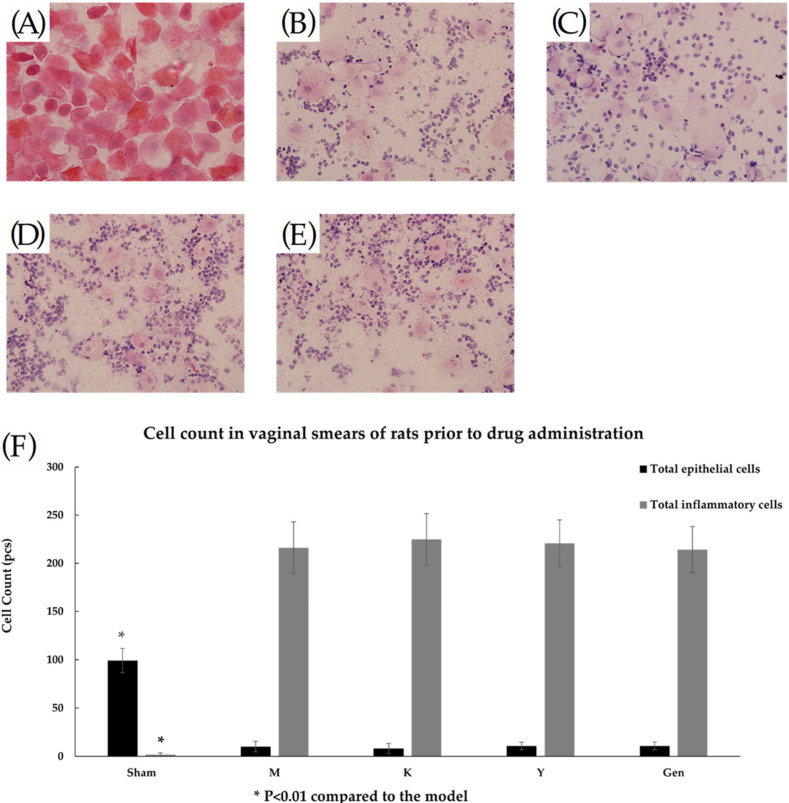
(**A**) After HE staining treatment, a vaginal smear from the Sham group of rats before administration of the drug showed a large number of epithelial cells and a very small number of inflammatory cells, (40Χ). (**B**) After HE staining treatment, vaginal smears from rats in group M before dosing were heavily infiltrated with inflammatory cells and a few epithelial cells, (40Χ). (**C**) After HE staining treatment, vaginal smears from rats in group K before dosing were heavily infiltrated with inflammatory cells and a few epithelial cells, (40Χ). (**D**) Vaginal smears from rats in group Y after treatment with HE stain, (40Χ). (**E**) Vaginal smears from rats in group Gen after treatment with HE stain, (40Χ). (**F**) Results of separate counts of epithelial cells and inflammatory cells in vaginal smears of rats before administration. * *p* < 0.01 compared to the model group.

**Figure 4 molecules-27-02388-f004:**
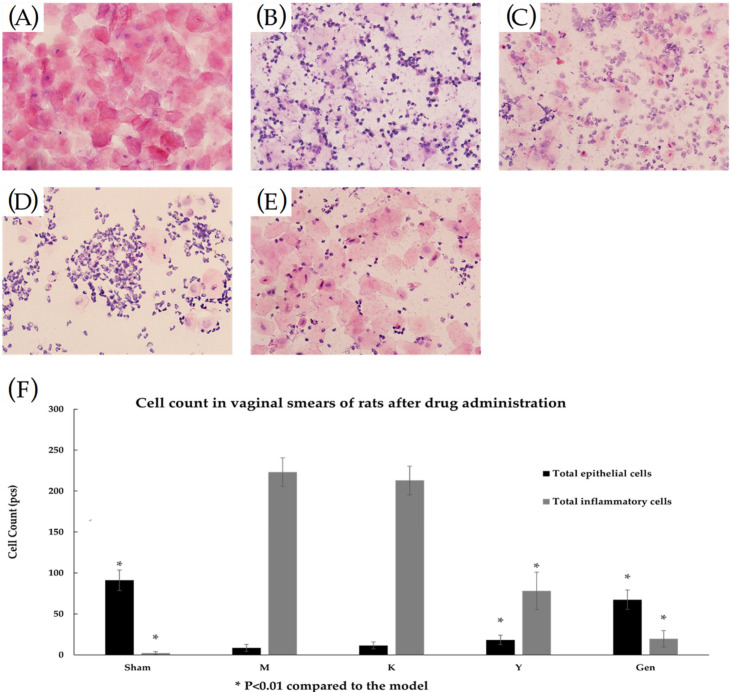
(**A**) After HE staining treatment, a vaginal smear from the Sham group of rats after administration of the drug showed a large number of epithelial cells and a very small number of inflammatory cells, (40Χ). (**B**) After HE staining treatment, vaginal smears from rats in group M after dosing were heavily infiltrated with inflammatory cells and a few epithelial cells, (40Χ). (**C**) Vaginal smears from rats in group K after treatment with HE stain, (40Χ). (**D**) After HE staining treatment, vaginal smears from rats in group Y before dosing were heavily infiltrated with inflammatory cells and a few epithelial cells, (40Χ). (**E**) After HE staining treatment, vaginal smears from rats in group Gen before dosing were heavily infiltrated with inflammatory cells and a few epithelial cells, (40Χ). (**F**) Results of separate counts of epithelial cells and inflammatory cells in vaginal smears of rats after dosing. For the Y and Gen groups compared to the M group, the number of epithelial cells increased significantly and the number of inflammatory cells decreased significantly, with the Gen group having a more significant effect. * *p* < 0.01 compared to the model group.

**Figure 5 molecules-27-02388-f005:**
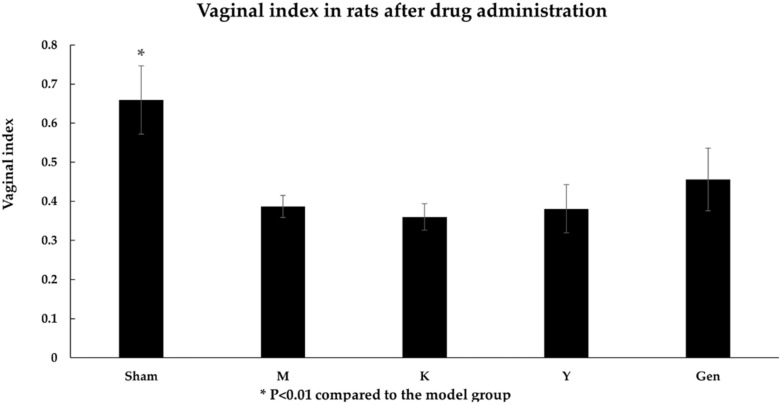
The vaginal index of the rats was significantly lower in the M group than in the Sham group and improved in the Gen group. * *p* < 0.01 compared to the model group.

**Figure 6 molecules-27-02388-f006:**
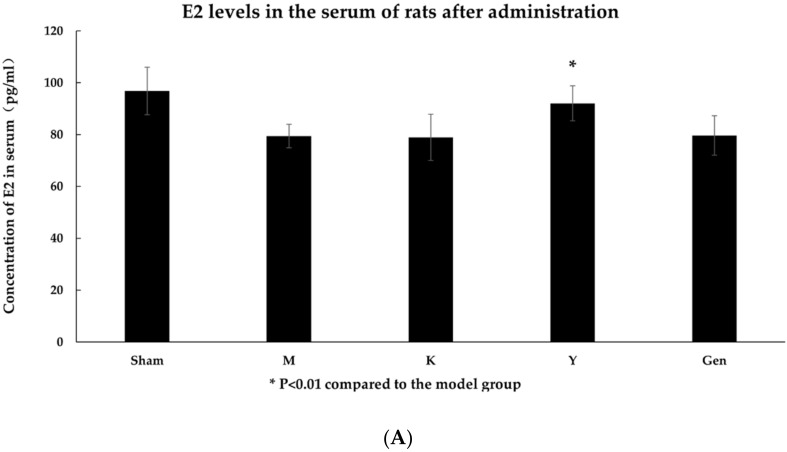
(**A**) The concentration of E2 in the serum of rats after administration was significantly higher in Sham and Y groups than in M group. (**B**) The concentration of FSH in the serum of rats after administration was significantly lower in the Sham and Y groups than in the M group. * *p* < 0.01 compared to the model group. ^#^
*p* < 0.05 compared to the model group.

**Figure 7 molecules-27-02388-f007:**
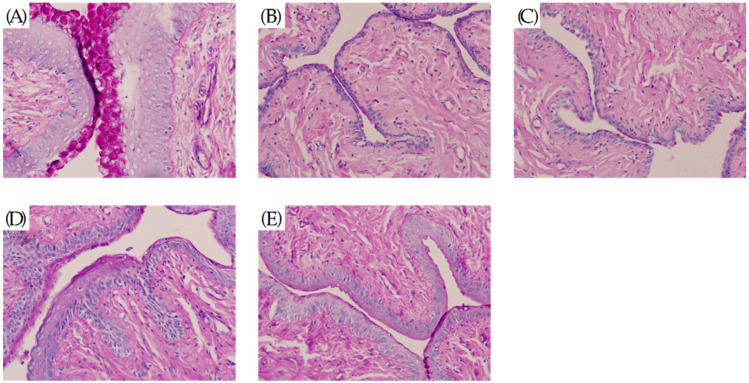
(**A**) Vaginal tissue from rats stained by PAS (Periodic Acid-Schiff’s reaction) showed glycogen enrichment in the Sham group, (40Χ). (**B**) Vaginal tissues from rats stained by PAS showed minimal glycogen content in the M group, (40Χ). (**C**) Vaginal tissues of rats stained with PAS showed little glycogen in the K group, which was not significantly different from the M group, (40Χ). (**D**) Vaginal tissues of rats stained with PAS showed increased glycogen content in group Y compared to group M, (40Χ). (**E**) Vaginal tissues of rats stained with PAS showed increased glycogen content in group Gen compared to group M, (40Χ).

**Figure 8 molecules-27-02388-f008:**
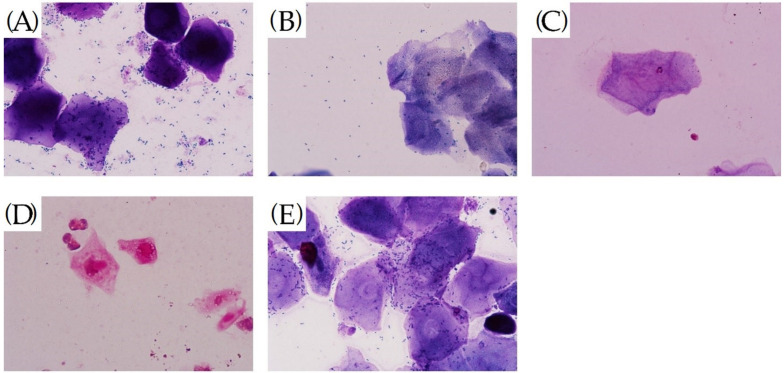
(**A**) The vaginal secretions of Sham rats were stained with Gram stain that showed a high level of lactic acid bacteria, (100Χ). (**B**) Vaginal secretions from rats in group M were stained with Gram stain that showed only a small amount of lactic acid bacteria, (100Χ). (**C**) Vaginal secretions from rats in group K were stained by Gram stain that showed only very low levels of lactic acid bacteria, (100Χ). (**D**) Vaginal secretions from rats in group Y were stained with Gram stain that showed only a small amount of lactic acid bacteria, (100Χ). (**E**) The vaginal secretions of Sham rats were stained with Gram stain that showed a high level of lactic acid bacteria, (100Χ).

**Figure 9 molecules-27-02388-f009:**
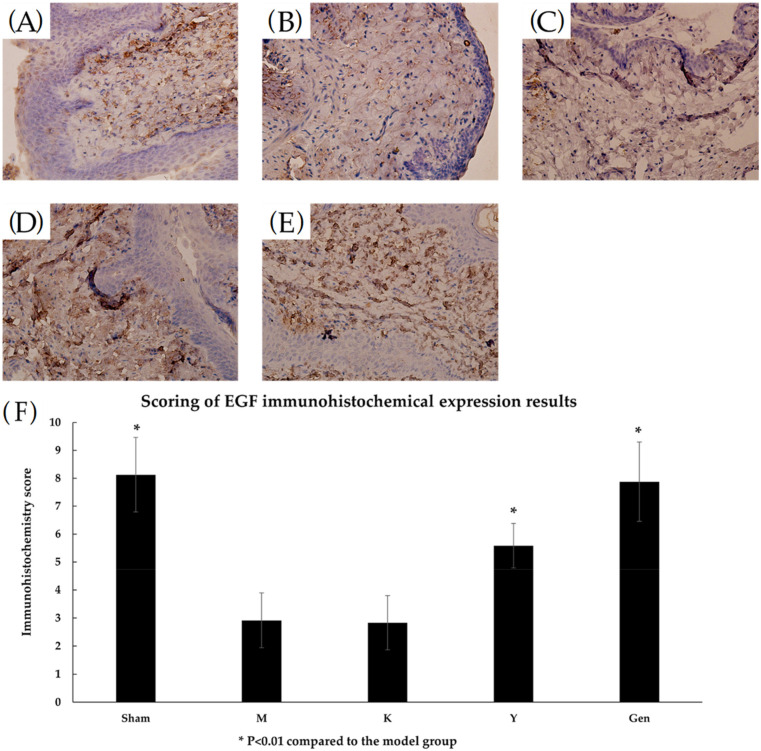
(**A**) High expression of EGF protein in the vaginal tissues of Sham rats, (40Χ). (**B**) Low expression of EGF protein in vaginal tissues of group M rats, (40Χ). (**C**) Low expression of EGF protein in vaginal tissues of group K rats, (40Χ). (**D**) Immunohistochemical results of vaginal tissues of rats in group Y with moderate positive expression of EGF protein, (40Χ). (**E**) Immunohistochemical results of vaginal tissues from the Gen group of rats were strongly positive for EGF protein with high expression, (40Χ). (**F**) EGF immunohistochemistry score results for each group. * *p* < 0.01 compared to the model group.

**Figure 10 molecules-27-02388-f010:**
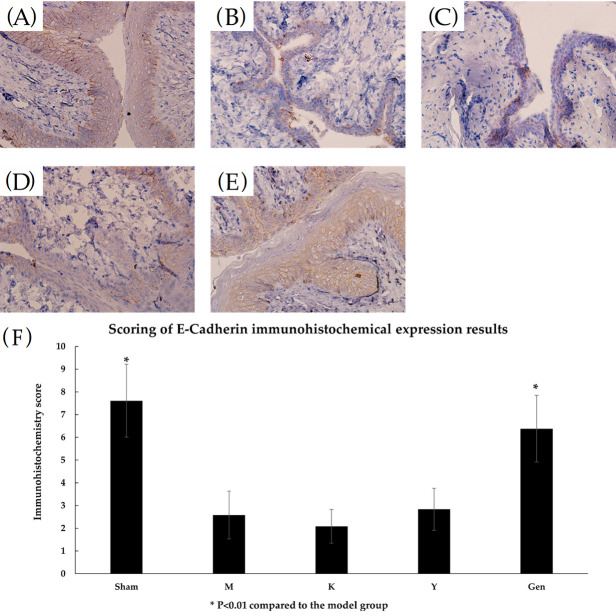
(**A**) High expression of E-cadherin protein in the vaginal tissues of Sham rats, (40Χ). (**B**) Low expression of E-cadherin protein in vaginal tissues of group M rats, (40Χ). (**C**) Low expression of E-cadherin protein in vaginal tissues of group K rats, (40Χ). (**D**) Immunohistochemical results of vaginal tissues of rats in group Y with low positive expression of E-cadherin protein, (40Χ). (**E**) Immunohistochemical results of vaginal tissues from the Gen group of rats were strongly positive for E-cadherin protein with high expression, (40Χ). (**F**) E-cadherin immunohistochemistry score results for each group. * *p* < 0.01 compared to the model group.

**Table 1 molecules-27-02388-t001:** Weight change before and after administration.

	Before Administration (g)	After Administration (g)
Sham	295 ± 9.82 *	314.13 ± 10.47 *
M	355 ± 10.81	381.5 ± 10.97
K	356.75 ± 8.80	381 ± 10.46
Y	361.5 ± 6.73	378.38 ± 7.26
Gen	352.375 ± 8.20	372.625 ± 8.50

* *p* < 0.01 compared to the model group.

**Table 2 molecules-27-02388-t002:** Relative mRNA expression of EGF, E-cadherin.

	EGF	E-cadherin
Sham	419.44 ± 46.08 *	69.72 ± 10.33 *
M	1	1
K	0.02 ± 0.01 *	4.39 ± 1.63 #
Y	8.74 ± 0.39 *	11.57 ± 3.33 *
Gen	710.28 ± 156.45 *	404.62 ± 36.21 *

* *p* < 0.01 compared to the model group. # *p* < 0.05 compared to the model group.

## Data Availability

The data presented in this study are available on request from the corresponding author.

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
