# Peer review of "Genistein Up-Regulates the Expression of EGF and E-Cadherin in the Treatment of Senile Vaginitis"

_molecules, 2022, doi:10.3390/molecules27082388_

Round 1

Reviewer 1 Report

This is an interesting study, and the authors have shown that genistein can promote the proliferation of vaginal epithelial cells by significantly increasing the expression of EGF and E-cadherin in an ovariectomized rat model. However, in my opinion, the paper is not well written and structured, and the English language is not good enough for publication. The paper has some shortcomings in regards to some data analyses and text, and I feel this unique dataset has not been utilized to its full extent. Below, I have provided numerous remarks on the text. Given these shortcomings, the manuscript requires major revisions.

  • This paper needs much work because it is not written in good English and cannot be accepted in its current form.
  • In-text citation: the reference does not meet the format requirements of Molecules.
  • P3, Table 1: Missing SI unit for weight changes of the rats.
  • Poor quality figures (Figure 1A–B, Figure 3F, Figure 5, Figure 6A–B, Figure 9F, Figure 10F)
  • SI units should be used. The scripted letter "u" is not an approved symbol for the liter.
  • P2, L54–58: It can bind to estrogen receptors…, but does not cause stimulation. Please clarify the sentence. Additionally, I suggest a major rewrite of this paragraph (Genistein…and postmenopausal women [4]). It should highlight the significance of genistein's recent advances in its ability to improve estrogen deficiency symptoms and the importance of further research.
  • P12, L350–352: This is a very vague statement. The manuscript does not provide any information concerning the analytical instrument used.
  • P13, Item 4.6: The authors should provide information on the anesthetic drug used in surgical procedures.
  • P12, L325–334: This section should be discussed in-depth detail in conjunction with the relevant studies.
  • P12, L335: Because genistein is a well-studied phytoestrogen, the authors should emphasize the uniqueness of this study and its significance.

Author Response

I am very honoured to have your review and I apologise for any errors in my paper that may have caused you problems. I have made the relevant changes to the issues you have pointed out, please see the attached document for details. I hope that this paper will be well received by you. I wish you a happy life.

Reviewer 2 Report

This is a well written research experimental article investigating the therapeutic mechanism of genistein (Gen) on postmenopausal senile vaginitis. Introduction is good. The last statement of introduction (last paragraph) is the aim (l. 76-78) I would prefer that you state it as a hypothesis of your research (e.g., that indeed there are therapeutic effects) and to explore them further. 

Results are presented very nice. However several figures need be replaced with ones of better quality (Figures 1 , 3, 4 , 5) the links you provided on the images are not working. Histopathological figures should be accompanied with a scale bar (μm - micrometers), however, since you provide the magnification, it is not totally necessary. Discussion should also include implications for further studies. 

Author Response

(The authors gave the same response as above.)

Round 2

Reviewer 1 Report

The reviewer thanks the authors for the review work done and the attention given to the remarks made. Unfortunately, the authors did not consider some recommendations realized in the last revision. There are also many types of errors. I see many spacing errors between words, and I have indicated those errors as minor comments. Those errors are too many to list in my review. I hope the authors will spend some time correcting those errors. With this and other concerns, I have indicated below that I cannot recommend accepting the paper for publication in its present form.

Major concerns

  • The only thing I would be careful of is consistency in writing. It's not a good idea to mix US and UK spelling, so I think it's better to either follow UK spelling rules or US.
  • Page 2, line numbers 47 – 49: The statement needs a reference.
  • Page 2, line number 96: Please revise the words ‘(10-8-10-6 M)’ and ‘(>10-5 M)’.
  • Page 2, line numbers 96 – 98: Please consider checking the chemical name, ‘dynein’. I think that it should be genistein, not dynein. This is because dynein is a family of cytoskeletal motor proteins that move along microtubules in cells. They convert the chemical energy stored in ATP to mechanical work and it does not involve this context.
  • Page 3, line numbers 64 – 66: Please consider moving the objective of the study to the final paragraph of the introduction section.
  • Page 14, line numbers 401 – 402: The authors indicated that they have evaluated the effects of the genistein prepared in a suppository dosage form. However, the authors did not clearly indicate the exact percentage of genistein incorporated into the test formulation. It is not possible to say for certain that the prepared suppositories contained 100% genistein. The authors should specify the exact concentration of genistein in the finished product.

Minor concerns

  • Page 1, line number 20: Space between ‘…, respectively.Geninsitein…’
  • Page 1, line number 42: Space between ‘…inflammation[1]’.
  • Page 2, line number 54: Delete ‘.’ before the word ‘The degeneration’.
  • Page 2, line number 56: Space between ‘…).Estrogen’.
  • Page 2, line number 61: Instead ‘ph’ with ‘pH’.
  • Page 2, line number 61: Instead ‘Susceptible’ with ‘susceptible’.
  • Page 3, line number 101: Consider adding ‘.’ after the word ‘cancer’.
  • Page 11: the word 'Figure' in the manuscript is written in two ways, 'Figure' and 'Fig.', the author should consider only one format to ensure consistency of the article.
  • Page 11, line numbers 304 and 307: space between ‘…,(40X)’
  • Page 11, line number: 307: delete ‘.’ after the word ‘protein’
  • Page 12, line number 308: space between ‘expression,(40X)’.
  • Page 12, line number 316: space between ‘positive.Group’.
  • Page 13, line number 356: instead ‘PH’ with ‘pH’ and ‘Glycogen’ with ‘glycogen’.
  • Page 13, line number 369: instead ‘mole-cule’ with ‘melecule’
  • Page 14, line number 375: space between ‘wall.RT-PCR’
  • Page 14, line number 403: instead ‘Detection method’ with ‘Analytical method’.
  • Page 14, line number 403: instead ‘mthanol’ with ‘methanol’.
  • Page 14, line number 404: space between ‘1.0ml/min’
  • Page 14, line number 404: instead ‘ul’ with ‘ml’
  • Page 14, line number 405: space between ‘262nm’
  • Page 14, line number 406: space between ‘200(Zhejiang…’
  • Page 14, line number 408: space between ‘type(Shanghai…’ and ‘…800KDB(Kunshan…’
  • Page 14, line number 410: space between ‘dosing:Suppositories’
  • Page 14, line numbers 415 – 417: space between ‘Kit(Thermo’, ‘kits(Cwbio)’, ‘kit(Beijing…’, ‘buffer(Solarbio)’, and ‘stain(Nanjing…’
  • Page 14, line number 422: space between ‘Labeler(TECAN …’
  • Page 14, line number 422: space between ‘Microscope(Olympus…’
  • Page 15, line number 429: space between ‘300g’
  • Page 15, line number 429: Instead ‘…selected, Animals were…’ with ‘…selected. Animals were …’
  • Page 15, line number 473: space between ‘formula:Organ’
  • Page 16, line number 484: Instead ‘5um’ with ‘5 mm’
  • Page 16, line numbers 511 – 512: Degree measures of temperature are normally expressed with the ‘℃’, not ‘°’ symbol’.
  • Page 16, line number 513: Instead ‘m RNA’ with ‘mRNA’.

Author Response

I'm sorry that my last edit didn't meet your requirements, and my lack of work has added to your workload, for which I'm very sorry. I also want to thank you again for your comments, they will make my article better and will help me a lot in future articles. Therefore, I have tried my best to revise every correction you have made, and have thoroughly reviewed and revised the issues you did not list. I am also looking for more professional help with language issues. I hope this modification meets your requirements and fulfills your requirements.

Thanks again for your work and I hope to hear from you that the review has passed. 
